# Time preferences and their life outcome correlates: Evidence from a representative survey

Dániel Horn[1,2☯], Hubert János Kiss[1,2☯]*

**1** Center for Economic and Regional Studies, Institute of Economics (KRTK KTI), Budapest, Hungary,
**2** Corvinus University of Budapest, Budapest, Hungary

☯ These authors contributed equally to this work.
* kiss.hubert.janos@krtk.mta.hu

## Abstract

We collect data on time preferences of a representative sample of the Hungarian adult population in a non-incentivized way and investigate how patience and present bias associate with important life outcomes in five domains: i) educational attainment, ii) unemployment, iii) income and wealth, iv) financial decisions and difficulties, and v) health. Based on the literature, we formulate the broad hypotheses that patience relates positively, while present bias associates negatively with positive outcomes in the domains under study. With the exception of unemployment, we document a consistent and often significant positive relationship between patience and the corresponding domain, with the strongest associations in educational attainment, wealth and financial decisions. We find that present bias associates significantly with saving decisions and financial difficulties.

## Introduction

Economic analysis is based on the tenet that individuals maximize utility functions that are representations of preferences, given some constraints. One of the most basic preferences that appear in almost all introductory books on Economics are *time preferences*. There are two relevant aspects of time preferences that we study in this paper. On the one hand, patience reveals how an individual values the future relative to the present, hence it plays an essential role in intertemporal decision-making. Importantly, patient individuals have low discount rates, so they appreciate future benefits more than their more impatient peers. Thus, as the cost of effort generally materializes in the present, more patient individuals are expected to invest more today in things that bear fruit tomorrow. As a consequence, more patient individuals may tend

- to invest more in their human capital by studying more;
- to choose long-term financial investments (e.g. retirement savings);
- to lead a healthier life.

**Data Availability Statement:** All relevant data are within the manuscript and its Supporting Information files.

**Funding:** This study was funded by the following: DH - K 124396, National Research, Development & Innovation (NKFIH) HJK - K 119683, the National

Research, Development & Innovation (NKFIH) HJK - 109354, Hungarian Scientific Research Fund (OTKA) HJK - ECO2017-82449-P, Spanish Ministry of Economy, Industry and Competitiveness HJK - Higher Education Institutional Excellence Program of the Ministry for Innovation and Technology in the framework of the „Financial and Public Services" research project (reference number: NKFIH-1163-10/2019) at Corvinus University of Budapest The funders had no role in study design, data collection and analysis, decision to publish, or preparation of the manuscript.

**Competing interests:** The authors have declared that no competing interests exist.

Note that these accumulation decisions affect the physical / human capital and factor productivity in the society that in turn are direct factors determining the income of the country. There is an important literature [1–4] that studies how different determinants, such as patience, affect macroeconomic growth.

The other aspect of time preference is time consistency. Time-consistent individuals have a constant discount rate between any two equidistant points in time. However, a large share of individuals are not time-consistent, but have a higher immediate discount rate relative to their long-run discount rate. In other words, many individuals are more impatient in the short run than in the long run. These present-biased individuals place excessive weight on immediate costs compared to benefits in the future. This tendency may lead to procrastination, because even if we want to achieve a goal (e.g. save more money or lose weight), given that we perceive the immediate costs to be too high, we may want to delay incurring those costs. Therefore, more present-biased individuals may have a tendency, for instance, to underinvest in educational attainment or to have financial problems.

There is an active empirical literature that attempts to measure individual time preferences and link those preferences to individual decisions. This paper contributes to this literature. We elicited time preferences in a survey in Hungary. Our data have at least three desirable features. First, the survey is representative of the adult population of a whole country, which is still not very common of papers studying individual preferences, notable exceptions being [5, 6]. Second, it provides a very rich set of controls (including risk attitudes), which allows us to see if our preference measures have predictive power once we control for a wide range of variables. In fact, one of the contributions of the paper is to see how the association between our preference measures and the life outcomes changes as we add more and more controls. Third, similar studies have been carried out, but mostly in the US and Western countries. Less is known about other, in our case Central-Eastern European, countries.

We link our time preference measures to life outcomes in five domains: i) educational attainment, ii) unemployment, iii) income and wealth, iv) financial decisions and difficulties, and v) health. In general, we hypothesize that more patient individuals fare better in these domains (e.g. have better educational attainment or higher income), while present bias may lead to worse outcomes in these domains, *ceteris paribus*. We find that with the exception of unemployment, patience is indeed associated with the outcomes that we investigate in the expected way. Moreover, these associations are often significant even if we control for a host of variables. For instance, we can say if patience is associated with wealth once we take into account education (that may mediate the effect of patience on wealth). We document the strongest relationships in escaping low educational attainment, wealth and financial decisions. Present bias often exhibits no consistent relationship, though we report strong associations in the expected direction in financial decisions and financial difficulties.

Our findings complement the existing literature in the following ways. First, we investigate if results, found on non-representative samples, hold for our representative sample as well. For instance, [7] find that present-biased individuals are more likely to have excessive credit card borrowings, but their sample is by no means representative of the US population. The same is true for [8], who study the predictive power of time preference on a special educational performance using a non-student population, but one may have doubts to which extent truck drivers' choices reflect those of the wider population. Second, we study outcomes that are not investigated in other papers that use representative samples. Most of the papers focus only on some life outcomes, while we attempt to gain a general insight on the association between time preferences and the important life outcomes. For example, one of the papers that is closest to ours is [5] that also uses a representative (and incentivized) sample to see how time preference correlates with health, energy use and financial outcomes. Our scope of life outcomes is wider,

as we also study educational attainment and unemployment (but we do not consider energy consumption). Third, we are among the few (along with [7] and [5]) that measure both patience *and* present bias (for instance, [6] have a global, representative survey, but they do not have data on present bias).

Since we use cross-sectional data both for time preference and the life outcomes, we can only speak about correlations between the variables. Hence, we cannot claim that patience causes the outcomes, the causality may run the other way around (e.g. education affecting patience). However, the fact that we study five life outcomes allows us to see if the separate life outcomes associate with time preferences even if we control for the rest of life outcomes. Hence, we do not simply establish if time preferences relate to saving decisions, but we are able to tell if there is an association even after considering educational attainment, income and other outcomes.

The rest of the paper is structured as follows. Next, we present the data and the hypotheses. Then, we show our main results. The paper concludes with a discussion.

## Data and hypotheses

In this section we present the data, and based on the literature we formulate hypotheses on the possible associations between our time preference measures and life outcomes in the five domains. We also discuss some issues related to time preferences.

### Data and preference tasks

The TÁRKI Social Research Institute carries out a quarterly survey in Hungary called Omnibus with a randomized sample of about 1000 individuals aged 18 and over. The survey is based on personal interviews, applies random selection sampling, and importantly it is representative of the Hungarian adult population in terms of gender, age, education and settlement type. The Omnibus survey has a fixed part that is asked in each quarter. It comprises the following socio-demographic data: gender, age, family status and structure, level of education, labor market status, individual and family incomes, wealth and financial situation, social status, religiosity. This gives a fairly comprehensive picture of the respondent along many relevant dimensions. In addition, researchers can add questions to the survey at a cost. TÁRKI charges for the new questions based on the supposed time it takes to answer them.

We introduced three items into the survey of January, 2017. For the exact wording of all three items see S1 Appendix. The first item included 5 questions regarding time preference that served to find the approximate indifference point of the surveyed individuals between an earlier and a later amount of money. Similarly to [6] we used the staircase (or unfolding brackets) method (see [9]) as it efficiently utilizes the available number of questions to approximate the indifference point between a present and a future payoff. The questions are interdependent hypothetical binary choices between 10,000 Forints (about 30.1 EUR / 33.2 USD at the time of the survey) today or X Forints in a month. The 10,000 Forints remain constant during the 5 questions while the amount X is changed systematically depending on the previous answers. For instance, if an individual prefers 10,000 Forints today to X = 15,500 Forints in a month, then it indicates that their indifference point is higher than 15,500 Forints, hence in the next question X is increased. There are $2^5 = 32$ possible outcomes corresponding to the last choice, which is our proxy for the indifference point. S2 Appendix contains the whole structure of the staircase method with the numbers that we used.

The second item measured risk attitude with a simple question that asked how much of 10,000 Forints the individual would place as a bet in the following gamble. We followed [10] when choosing this task. A bag contains 10 black and 10 red balls and one is drawn randomly.

The individual can choose a colour (black or red) and if the colour of the ball drawn coincides with their chosen colour, then she wins the double of the bet she placed. If the colour of the ball drawn from the bag is different from the chosen colour, then the bet is lost. We explained also that the amount not placed as a bet would be given hypothetically to the individual. We regard the amount placed as bet as a natural measure of risk attitude. Our measure is very similar to the investment game of [11]. [12] compare four experimental risk elicitation methods, among them the investment game. There is no clear best elicitation method, different methods have different advantages and shortcomings. Importantly, a message of the study is that all the methods are useful to distinguish individuals according to their risk preferences.

The third item also measured time preference and was almost identical to the first one, but the time horizon was different as the earlier hypothetical payoff occurred in a year, while the later one in a year and a month. Note that the distance between the payoffs (1 month) is the same as in the first item. The order of the preference tasks was the same as described here. As only the risk attitude measure separated the time preference items, possibly there were individuals who strived to be consistent (that is, make the same choices), so potentially we underestimate the extent of present bias in our sample.

## Five domains and the overarching hypotheses

We examine five domains where we expect to see correlations between time preferences and outcomes. Our data do not allow us to establish causal relationships, so we limit ourselves to speak about associations between our preference measures and the life outcomes on an individual level. In contrast to our paper, [13] investigate how time preference associates with innovation, environmental protection, credit rating and body mass index at country level. Here, we review briefly the relevant literature that often goes beyond associations and investigates the underlying mechanisms as well.

**Educational attainment.**   There are many studies that show that non-cognitive skills—and among these skills many related to time preferences—are important determinants of educational attainment [14–17]. [18] use longitudinal data from Sweden that links measured time preference at age 13 to schooling outcomes gained from administrative registers. Their results indicate that high discount rates (that is, less patience) are related to worse school performance and educational attainment. [8, 19–22] report similar findings in different countries and settings. [6] find that this result is valid when considering representative data from 76 countries. However, [23] fails to find correlation between patience and school performance in a dataset from the US. Using data from Vietnam, [24] also find that more patient individuals achieve higher educational levels, but they do not find a relation between present bias and educational attainment. The presence of self-control problems and procrastination (both tightly related to present bias) in an educational context has been shown in many studies, see for instance [25–27]. Negative association of present bias with school performance is documented in several papers (e.g. [28–30]). The importance of these findings stems from the fact that education outcomes determine success in life to a large extent as captured, for instance by the wage premium (e.g. [31]) or the positive relation between schooling and other socioeconomic outcomes (for example [32]). Overall, the evidence in the literature suggests that there is a positive relationship between patience and educational attainment. Even though the evidence seems weaker, we also expect to find a negative association between present bias and educational attainment.

In our dataset, we have information on the respondents' education level. Respondents place themselves into 9 categories: 1) lower than primary (1%), 2) primary (16%), 3) vocational secondary without maturity exam (31%), 4) vocational secondary with maturity exam (18%), 5) secondary with maturity exam (14%), 6) non-tertiary vocational qualification (5%), 7) tertiary

vocational qualification (1%), 8) tertiary BA level (11%), 9) tertiary MA level (3%). We define two outcome variables. The first one is a dummy that indicates if an individual was *not* able to get the maturity (school-leaving or A-levels) exam. The second one is also a dummy revealing if an individual has a tertiary level degree or not. Hence, our first education variable informs if an individual is not able to escape low educational attainment, while the second one captures if an individual reaches the highest level of schooling. Based on the previous hypotheses, we expect to see that more patient individuals tend to be less (more) likely to end up with a low (high) level of education. We conjecture that present bias is positively (negatively) associated with a low (high) level of education. We have also run our models with different specifications. We tested a linear specification (years spent in education) as well as a 5-category variable (primary, secondary with no-maturity exam, secondary with maturity exam, secondary with maturity and vocational qualification, tertiary). Results show that differences are clearest between those with and without maturity exam. As dependent variable we use a 5-category education variable (primary, secondary with no-maturity exam, secondary with maturity exam, secondary with maturity and vocational qualification, tertiary) to retain more information in our models.

**Unemployment.** There are various examples that show that time preference associates with unemployment. [33] find that low self-control in childhood and adolescence predicts youth unemployment in New Zealand. [34] show that truck driver trainees who were more impatient or present-biased were more likely to quit the training program in the US. [18] report that patience is negatively related to unemployment using data from Sweden. Using British data, [35] find that children that had low self-control were more likely to be unemployed as an adult. Using a Dutch longitudinal survey, [36] document that both on-the-job search and work effort increase with patience which may imply a lower threat of unemployment. In the database we have data on the employment status of the respondents. We consider only individuals who are active on the labor market, so we disregard students, retired and other inactive individuals and we test if unemployment is negatively (positively) associated with patience (present bias), *ceteris paribus*. Of the 998 respondents, 639 are employed or self-employed, 269 retired, 32 unemployed, 19 students, 22 on maternal leave, while the rest is other inactive.

**Income and wealth.** Time preference might also relate to income and wealth. [37] documents that in the US poor households exhibit less patience than rich ones. Similar findings have been reported for Ethiopian (see [38]) and South Indian (see [39]) households. Using data from Vietnam, [24] find that income and patience are positively correlated, however there is no correlation between present bias and income. [40] report that even after controlling for social class origin and IQ, self-control measured in childhood has predictive power for income earned as an adult in New Zealand. [18] document that patience is positively related to earnings and disposable income in Sweden. Studying low-income US households, [41] find evidence that scarce resources (low income and wealth) is able to affect the willingness to delay gratification: poor participants (in their study individuals before payday) were more present-biased than participants with more resources.

In our survey, there were two questions on the individual post-tax (net) income. Respondents could either report an estimated average monthly amount of their income (547 out of the 998 surveyed individuals did so) or could indicate the level of their income on an 8-level categorical variable (176 out of the 998 individuals did so). We have imputed the 8-category income variable with the estimated average monthly income variable for the controls, and conversely imputed the first income measure with the means of the categories, when we used income as the dependent variable. Thus, in all regressions below we will control for the individual income using dummy variables for the 8-category income (and an additional dummy

for the missing responses), while we use the estimated average monthly income measure for the analysis of income on time preferences.

To proxy wealth we have generated the principal factor of six dummy variables indicating whether the respondent has a 1) car, 2) dishwasher, 3) washing machine, 4) landline phone, and 5) whether the respondent owns the property she lives in, and whether 6) she owns another real estate property. We have replaced missing values (for 34 respondents of the total 998) on this principal factor with zero (the average value) and included a *missing* dummy to control for this in the regressions below.

Based on the literature, we conjecture that patience and income (and wealth) are positively correlated. Moreover, we expect a negative relationship between present bias and income (and wealth).

**Financial decisions and financial difficulties.** *Banking and saving decisions*. Related to the previous point, wealth may be due to saving decisions, so time preferences may influence those decisions as well. [10] find that impatience predicts saving decisions of Austrian adolescents (more impatient ones are less likely to save). [5] report a positive relationship between patience and different forms of savings (credit card balance, retirement and non-retirement savings) for a representative sample of the US adult population. [6] find that this finding holds globally. [42] study mortgage choices in the US and also the management of this obligation. They find that those who are more impatient and present-biased tend to choose mortgages that minimize up-front costs. Results are not unambiguous as [43] find no correlation between individual long-run discount factors and borrowing behavior. Interestingly, while impatience increases homeowners' willingness to abandon a mortgage, present bias does the opposite, showing that both aspects may be important in financial decisions. Interestingly, some studies [44, 45] do not find a significant relationship between patience and financial decisions. Moreover, some do find association between those decisions and present bias [30, 45]. The relation between self-control problems (a manifestation of present bias) and saving decisions have been also proposed [46, 47]. Evidence on this relationship is provided in numerous papers [48, 49]. The literature is suggestive that more patience is related to more saving, but present bias may hinder carrying out saving plans.

In our dataset, we have extensive data on financial decisions. More precisely, we know if an individual in the sample has a 1) debit / credit card, 2) owns shares, 3) has a bank account, 4) has retirement savings, 5) has life insurance. From these variables, we create an index variable meant to capture financial decisions in general. [44] suggest that it is more likely that we may find correlations between the preference measures and *aggregated* measures of behaviors (e.g. overall saving decisions), than single components (e.g. owning share or having retirement savings), because in the latter idiosyncratic effects may play a big role, while they tend to cancel out as more and more individual behaviors are added into an aggregate measure. Principal factor analysis yields two meaningful indices. The first has high factor loadings from variables 1 and 3 (whether the individual has a bank account and debit / credit card), while the second has higher positive factor loadings from variables 2, 4 and 5 (owning shares / having retirement savings / having life insurance), and negative factor loadings from 1 and 3. We call the first factor *banking decisions* and the second *saving decisions*. The banking decision index indicates if an individual has at least reached the lowest levels of financial inclusion. The second index points to higher levels of financial sophistication. We impute missing values with zero in both of these indices and include a dummy for the imputed values in the estimations below. Based on the literature, we expect that more patient individuals tend to i) use more the basic banking services, ii) save more. Regarding present bias, we expect to see the opposite relationships.

*Financial difficulties*. There is evidence that time preferences may be related to financial difficulties. [44] document that more patient individuals are more likely to pay their bills without

problems in a US sample. [7] report that present-biased individuals are more likely to have credit card debt and tend to have a significantly higher amount of debt, but patience does not affect credit card debt. [40] document that children in New Zealand, who had self-control problems, later (at the age of 32) were less likely to save and faced higher financial distress (e.g. money-management difficulties, credit problems). [34] show that credit scores correlate positively with patience using a special US sample (truck driver trainees). [50] find the same and show that impulsivity plays also a role. Using data from the US, [51] show that patience predicts creditworthiness. The evidence is not unequivocal. [52] find no correlation between patience, present bias and various forms of mortgage delinquency (e.g. being behind with the payment or missing part of the payment) in a US sample. [53] show that present bias is negatively correlated with an aggregate index of financial condition, a finding based also on a US sample.

Concerning financial difficulties, we know whether an individual has problems of 1) paying public utility bills, 2) servicing a mortgage or 3) servicing other types of loans. Using principal factor analysis, from these three dummy variables we create an index that proxies the extent of the individual financial difficulties. All three variables had similarly high positive factor loadings. Based on the extant literature, we expect more patience to be associated with less financial difficulties, while we conjecture the opposite relationship between present bias and financial difficulties.

**Self-reported health.**   Many associations have been established between time preferences and health issues. Regarding obesity, [18, 54–56] find that there may be an expected link between the two. [30, 57] show that not only patience, but also present bias matters for obesity in the US and Germany. [10, 58–61] argue that general measures of time preference and self-control are related to smoking behavior. [34] document that more impatient and present-biased individuals in their US sample are more likely to smoke, but they do not find any relation between the two aspects of time preference and the body mass index. [44] report that, in the US, more patient individuals are less likely to smoke, have a lower BMI and are more likely to exercise. [62] document that present-biased individuals with diabetes are less likely to follow clinical guidelines. [63] report that depression also affects time preference.

In more general terms, [5] find that both patience and present bias affect self-reported health and also different health behaviors (e.g. obesity, BMI, smoking, binge drinking) in the US. Analyzing data from New Zealand, [40] find that self-control problems in childhood predict health problems as adults. However, there are studies that report no evidence on the relationship between time preference and self-reported health [64].

In our survey, the respondents assess their health status on a 5-item scale ranging from very bad to very good. This is the only information on health. [5] has also a self-assessment question on health with a 5-item scale. Similarly to them, we form binary variables, the first showing if the individual has good self-reported health (the two top notches of the 5-item scale), while the second shows if the individual has bad self-reported health (the bottom two notches of the 5-item scale). Based on the literature, we expect a positive (negative) relationship between patience (present bias) and self-reported health.

Reviewing the literature in the five domains reveals that associations are not always clear and unambiguous. However, findings in the different domains point in the same direction. Overall, we expect patience to be positively related to positive outcomes in the domains that we consider, while present bias to have the opposite associations.

## Other variables in the survey

As mentioned earlier, the survey contains rich socio-demographic data. We have information on the gender, age and the marital status of the respondent, and we also know the number of

their children (if any). We know the region and the type of the settlement (village, township, city) where the respondent lives. We have information on the profession (e.g. skilled worker, agriculture, manager, housewife) and if she is employed in the public sector or not. We have also information on the interviewer's opinion about the race of the respondent. More precisely, the interviewer can indicate whether she thinks that the respondent was of Roma origin. (It is prohibited by law to ask the ethnicity of the respondent).

## Some considerations

Studying time preferences necessarily implies investigating the future. Since the future is inherently uncertain, risk attitudes may be confounded with time preferences. If we do not control for risk aversion, then we may underestimate the coefficient of time preferences. In fact, [65] and [66] report a significant negative correlation between individual discount factors and risk aversion, while [67] report the opposite association. However, [68] document that estimates of time and risk preference are independent. Moreover, [69] reject the hypothesis that risk and time preferences are governed by a single parameter and conclude that the relationship between the two is that individuals prefer to delay the resolution of risk. [70] also state that while risk and time preferences are intertwined, they are not different manifestations of the same phenomenon. [71] show that risk attitudes and time discounting are related through various channels, for example both risk tolerance and patience are on average higher for payoffs that materialize in the future compared to payoffs in the present. [72] provide evidence that time and risk preferences are different phenomena. [43, 73–75] argue also convincingly that risk attitudes should be taken into account when considering time preferences.

Importantly, if risk and time preferences are correlated, then by using only one of them may yield biased results. To alleviate this problem, when studying the associations between time preference and life outcomes we control for risk preference.

Another issue to keep in mind is domain specificity. While [76, 77] report similar discount rates for money and health, [78–81] found different rates, typically higher discount rates for money. [82] also document different discount rates for different domains (including for instance dating partners and cigarettes). However, [83] find no significant differences in the discounting of monetary and environmental outcomes, a result confirmed by [84]. [85] show that individuals discount more things that they find more tempting compared to those that are not so appealing to them. We cannot deal with this issue properly as we could insert in the survey only a restricted number of items and we opted for choice between monetary flows that is most widely used in the literature. For risk preferences [86] document that risk taking is domain-dependent. However, [87] report high correlation in risk taking in these different domains. Moreover, [12] show that domain-specific risk taking is often associated in a significant way with risk preferences elicited in experiments.

A related caveat concerns present bias. [88, 89] and [90], among others suggest that present bias may be more pronounced when considering consumption instead of monetary flows. Since we use monetary flows, we may underestimate present bias.

Similarly to [6] and differently from [5], we did not incentivize our preference questions. [91] find that the lack of incentives when measuring preference measures does not lead to different results than when the elicitation is incentivized.

The measurement of time preference generally involves choosing between an earlier and a later amount of money [89, 92]. If participants choose on different horizons, then we may calculate individual discount factors and comparing those individual discount factors may inform about whether an individual is present-biased or not. As a starting point, consider the model

of $(\beta, \delta)$-preferences proposed by [93] and [94]

$$U^t = u_t + \beta \sum_{s=t+1}^{\infty} \delta^{s-t} u_s, \tag{1}$$

where the overall utility $U$ in time $t$ is a function of hyperbolically discounted future utility, where discount rates decline as we move further away in time. Thus, the near future is discounted at a higher implicit discount rate than the distant future [94].

The amount to be received at the later time point in the longer-horizon time preference task can be seen as a proxy for the indifference point of an individual between receiving a fixed amount (in our case 10,000 Forints) earlier in time $t + s$ or a larger amount later at time $t + s + 1$ (in our case, in 12 vs. 13 months). We denote this indifference point by $x_{i,12-13}$, $12 - 13$ indicating that the decision is between an amount in 12 months and an amount in 13 months. Then, indifference implies

$$u_{12}(10,000) = \delta_{12-13} u(x_{i,12-13}). \tag{2}$$

This indifference point allows us to calculate the individual discount factor on this horizon ($\delta_{12-13}$) and we interpret it as a proxy for the individual's *patience*.

If the same choice is between now and some later date in the near future (in our case now vs. 1 month), then we can denote the indifference point by $x_{i,0-1}$ and the indifference relation becomes

$$u_0(10,000) = \beta \delta_{0-1} u(x_{i,0-1}). \tag{3}$$

Here, $\delta_{0-1}$ expresses the individual discount factor on the short run, while $\beta$ represents the degree of present bias that we will explain below.

## Results

### Descriptive statistics

In this section we investigate two things. First, we present the descriptive statistics of our time and risk preference measures and see if they correlate. Then, we study how preferences correlate with individual characteristics to understand which individual aspects are related to the preferences.

For time preference we follow [51], and we assume linear utility and calculate the individual discount factor, that is $\delta$, representing patience as

$$\delta = \frac{10,000}{x_{i,12-13}}. \tag{4}$$

The *higher* the individual discount factor, the more *patient* an individual is as she discounts the future less.

Since present bias expresses the idea that individuals are more impatient now than in the future, it means that $\delta_{12-13} > \delta_{0-1}$ and it is captured by $\beta$. Hence, $\delta_{0-1} = \beta * \delta_{12-13}$, that is

$$\beta = \frac{\delta_{0-1}}{\delta_{12-13}}. \tag{5}$$

Following the idea of [51] we use a dummy variable for present bias that takes on the value of 1 if $\beta < 1$. If $\beta = 1$ the person is time-consistent (our reference category). If $\beta > 1$ we will consider the respondent to be future-biased. Theoretically, the $(\beta, \delta)$-model is flawed to capture

**Table 1. Descriptive statistics: Patience and present bias.**

| Parameter | Average (St. Error) | Min | Max | Percentile | | | | | | |
|---|---|---|---|---|---|---|---|---|---|---|
| | | | | 5th | 10th | 25th | 50th | 75th | 90th | 95th |
| $\delta_{12\text{-}13}(N = 955)$ | 0.808 (0.160) | 0.465 | 1 | 0.465 | 0.541 | 0.714 | 0.833 | 0.952 | 1 | 1 |
| $\beta(N = 955)$ | 0.992 (0.185) | 0.465 | 2.15 | 0.663 | 0.788 | 0.943 | 1 | 1.023 | 1.162 | 1.305 |

future bias, as shown by [95]. However, following [7] we will use this definition of future bias in our analysis.

From the individual discount factors (shown in Table 1) we can calculate individual discount rates. The individual discount rate on the later time horizon (choices between amounts in a year and in a year and a month) has a median / mean value of 20% / 30.1%, with a wide range between 0% and 115% (standard deviation: 32.9%). [96] report a similarly wide range of individual discount rates in their survey. The corresponding numbers of the median and the mean for the earlier time horizon (choices between amounts now and in a month) are 22.5% and 34.3%. Using the classification of [51], 35.6% of our sample is present-biased, 28.1% is future-biased, and the rest (36.3%) is time-consistent. The share of present-biased individuals is close to those found by [7] (36%), [45] (about 27%) and [29] (30%) for non-representative samples from the US, Philippines and Hungary. Table 1 is similar in structure to table 3 in [5] and the distributions are comparable.

Our risk measure is the percentage of the endowment that the respondent places as a bet. Our task is very similar to the one used in [11]. [97] review numerous studies that use this task and find that participants tend to invest (bet in our case) about 50-70% of their endowment. We carried out the same task in a classroom experiment with university students and they risked on average 48.3% of their endowment [29]. In our current survey individuals on average risked 38.5% of their endowment, which is somewhat lower than levels found in the literature. This may be due to the fact that in most of those experiments university students were the participants, who are not representative of the population. There are studies [98, 99] that document a positive relation between cognitive abilities and risk taking. Since arguably university students have higher cognitive abilities than the representative population, this may explain why we observe a lower level of risk taking. Note also that people under age 40 and with a tertiary degree risk 54.5% of their endowment in our sample, as well. [100] study villagers in China and they report similarly low (and even lower) levels using a similar risk elicitation method.

Using pairwise correlation, we find that patience ($\delta$) and present bias ($\beta$) are negatively correlated (r = -0.38), that means that those who are less patient, are less likely to suffer from present bias in our sample. Note that this finding is somewhat artificial as a higher $\delta_{12-13}$ (our patience measure) implies that $\beta$ is smaller conditional on $\delta_{0-1}$. Note, however, as $\delta_{12-13}$ and $\delta_{0-1}$ are positively but not perfectly correlated (r = 0.66), the variance of $\beta$ is not zero. We also document a significant and positive correlation (r = 0.11) between patience and risk tolerance, suggesting that more patient individuals in our sample tend to be less risk averse. As mentioned earlier, [65] and [66] reported the same finding, but there are papers that fail to find such correlation. We detect some positive association between the present bias variable ($\beta < 1$) and risk tolerance, as present-biased individuals risk 6 percentage points (henceforth pp) more of their endowments (p-value = 0.004) compared to time-consistent people, but also future-biased individuals (when $\beta > 1$) risk 6 pp more than their time-consistent peers (p = 0.008).

To validate our measures, we briefly consider if we are able to find correlations reported in the literature in our data. A robust finding in the literature is that women are more risk averse

(see for instance [101] or [102]), especially when risk elicitation tasks as ours are used (see [97]). There are exceptions as for instance [103] do not find gender differences in risk aversion in a representative Danish sample. We find that women tend to risk approximately 6-9 pp less of their endowment than men, which is a sizable and significant difference. [87] uses a large, representative German survey and find that older individuals tend to be less risk taking, a finding that we share. However, findings are varied as [24] reports similar results as we do, however [103] document opposite findings. [104] using Dutch data report that more religious people are more risk averse. We find the same relationship in our data. Overall, we believe that our preference measures are meaningful.

## Regression analysis

We present our results using coefficient plots as these show in a clear way if a variable associates with the outcome of interest or not. The coefficient plots visualize the estimation of the coefficient with the corresponding standard errors. We relegate the corresponding OLS regression outputs to S3 Appendix.

In case of each domain, we estimate various models. In the first model we only add time preferences, that is patience ($\delta$) and the two dummies related to time inconsistency—present bias ($\beta < 1$) and future bias ($\beta > 1$). Note that if we employed only the present bias dummy and did not control for future bias, then we would compare present bias to time consistency *and* future bias. We believe that controlling for future bias allows us to gain a more accurate picture about the associations of present bias.

Then, in later models we add more and more controls. The first control is risk tolerance, measured by our preference task (see section Data and preference tasks). Then we include a set of variables that we call *exogeneous* controls. These include age, age squared, if the respondent is female, and if the interviewer believes that the respondent is of Roma origin. Using data from Spain, [105] show that Roma people may have different time preferences than the rest of the population has. The next set of control variables that we call *region* include dummies for the regions of Hungary and the type of settlement the respondent lives in. Regarding the regions, we have six dummies for the regions of Hungary, the baseline region being Central Hungary. We control for settlement type using three dummy variables (town, city, Budapest), the baseline being village. The fourth set of control variables are related to *family* and contains dummies related to the marital status (single, married, separated, living with partner, widow, divorced) and the number of children of the respondent. We have also control variables associated with *education* including dummies if the respondent has only basic education and if the respondent has a tertiary education degree. The set of control variables related to *income* contain information on the income level, on the wealth level (as detailed earlier in section Income and wealth) and on financial difficulties (see section Financial decisions and financial difficulties). The last set of control variables is related to *work* and has information on if the respondent works in the private / public sector and their employment status (e.g. unemployed, employee, employed, inactive etc.).

The sequence in which we presented our sets of control variables in the previous paragraph reflects the order as we include them into the regressions. Note that each new regression adds a new set of control variables to the ones that appeared in the previous regressions. Notice also that we will leave out the control variables that are directly related to the dependent variable. Thus, for instance, we do not control for *education* when we study which variables correlate with educational attainment. The coefficient plots visualize the associations at the 10 / 5% significance levels using thick / thin lines. S3 Appendix contains the coefficients and the significance of the variables of interest.

For the sake of completeness, in the figures we also show the coefficient of future bias. There is a growing literature that investigates future bias [95, 106–108]. We did not formulate any hypotheses and we will only comment on its association when it is noteworthy in our view.

## Educational attainment

Fig 1 shows the association of time preference with the probability of obtaining a university degree. Patience (delta) has the expected correlation with educational outcome: the more patient a respondent, the more likely that she has a diploma. Moreover, the association is marginally significant also when we control for risk attitudes. However, as we add more control variables, the relationship ceases to be significant. Note that the patience measure ($\delta$) in our sample ranges from 0.465 to 1. Hence, an interpretation of the coefficients is that individuals with a $\delta = 1$ are about 10 pp more likely to have a diploma than their peers with a $\delta = 0.5$ (if we restrict our attention to the significant coefficients). Present bias does not exhibit a significant association with the probability of obtaining a university degree in any of the specifications and does not even show a consistent pattern. For details, see Table A in S3 Appendix.

We carry out the same analysis at the other end of the education attainment distribution. That is, we investigate if time preference is different for people without the maturity exam. Interestingly, Fig 2 is not the mirror image of Fig 1, because we see that patience is not only significant when considering alone or with risk attitudes, but also as we add other control variables. The level of significance decreases gradually, but even after including all the control variables, patience is marginally significant at the 10% level. That is, even after taking into account a wide range of variables the more patient an individual, the less probable it is that she dropped out from education early. More concretely, individuals with a $\delta = 1$ are about 9.5 − 20 pp more likely to have at least a maturity exam than their peers with a $\delta = 0.5$. The

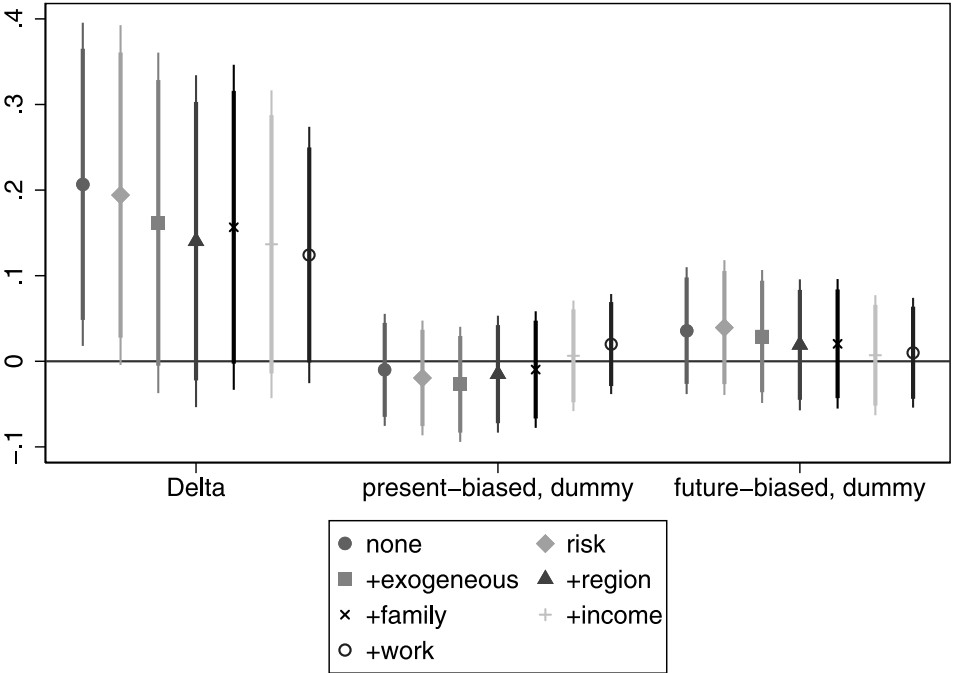

**Fig 1. The association of time preference and the probability of obtaining a tertiary degree.**

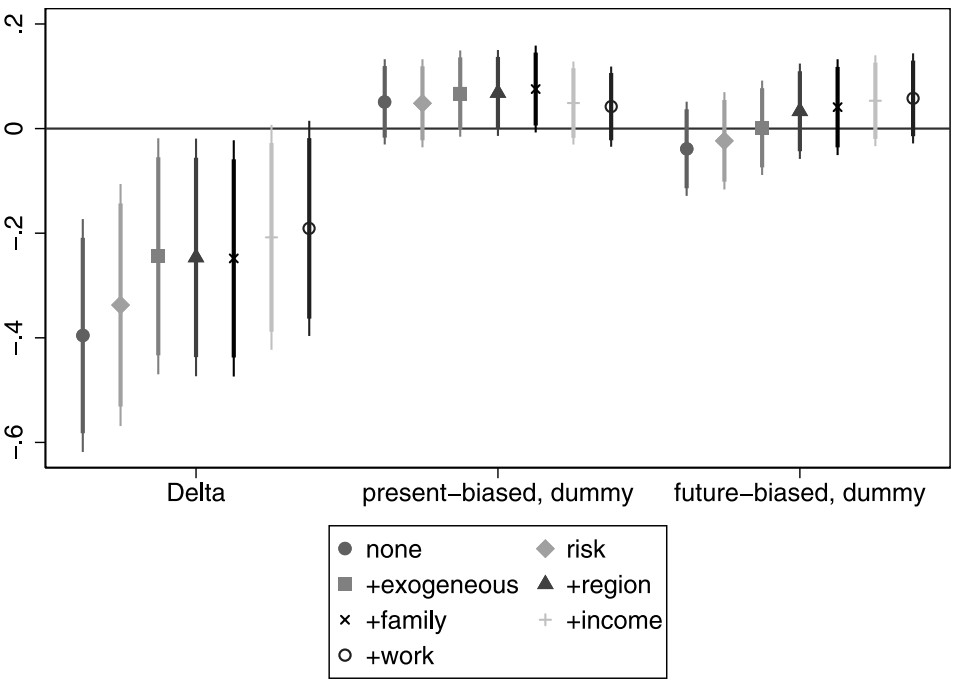

**Fig 2. The relationship of time preference with the probability of having no-maturity exam.**

asymmetric relationship of patience with educational outcomes suggests that it has a larger role in escaping low educational attainment than to obtain a tertiary degree.

The sign of the present bias dummy is also consistently above zero and, in some cases, it is even significant. In these cases, present-biased individuals are about 7.5 pp more likely to end up without a maturity exam. It loses significance when we control for income and work that are bad controls in the sense that they are highly correlated with educational attainment. This suggests that present-biased individuals are more likely to end up without a maturity exam, in line with our hypothesis. For details, see Table B in S3 Appendix.

**Finding 1 (education)**: Patience seems to be important in escaping low educational attainment even after controlling for a host of variables, but less important when considering attaining a diploma. Present bias seems to have a consistent positive association with having no maturity exam, but has a rather erratic relationship when investigating its role in obtaining a diploma.

## Unemployment

We hypothesize a positive (negative) relationship between patience (present bias) and being employed. We exclude from this analysis those individuals who are retired, on maternal leave or are students, because they are not active on the labor market.

Fig 3 shows that being employed is generally negatively correlated with patience and the association is often significant after we take into account the *risk*, *exogenous*, *region* and *family* controls. Surprisingly, we not only do not find any support for our hypothesis related to employment and patience, but we find the opposite. Present bias has an insignificant association with being employed. For details, see Table C in S3 Appendix.

We attempted various different specifications informed by the following arguments. One may claim that some individuals may prefer to be unemployed rather than having a job that

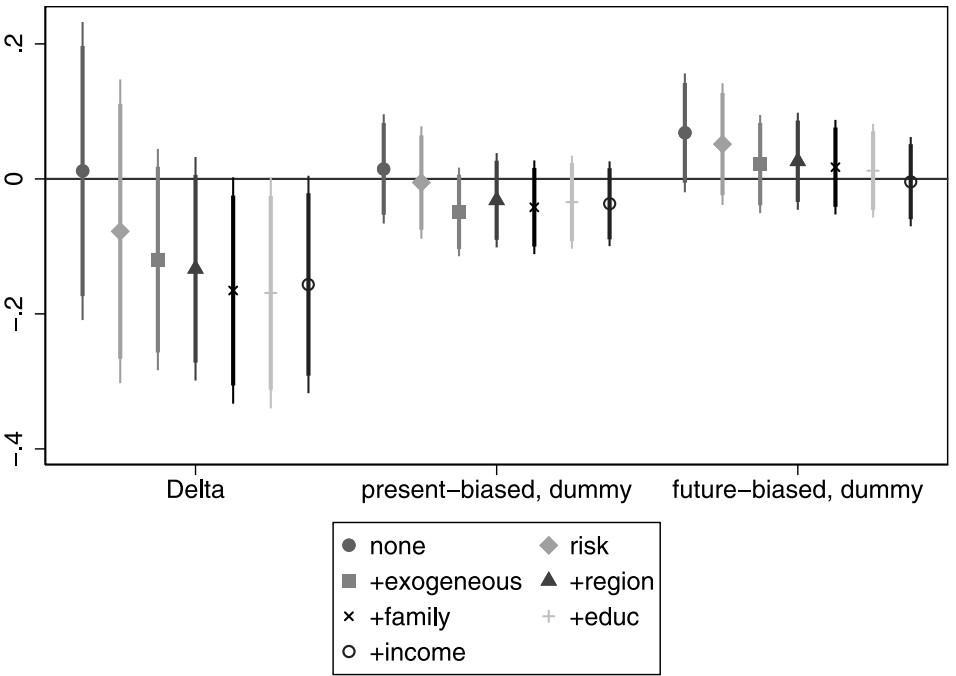

**Fig 3. The association of time preference with being employed.**

they dislike, so they possibly would wait to have an appealing offer. This may make unemployment spells longer and actually may make the correlation of patience with being employed weaker. Since potentially wealthy or well-off individuals could afford to wait, we excluded them from the analysis and reran the regressions for the respondents below the mean / median income. Though the numbers change somewhat, we see basically the same qualitative pattern. As a further check, we excluded the self-employed individuals from the analysis, but the results do not change here either.

**Finding 2 (employment)**: Patience is not associated with employment in the expected way. We observe that in many specifications more patient individuals are significantly less likely to be employed. Present bias often has the expected negative sign (present-biased individuals being less likely to be employed), but the association fails to be significant.

## Income and wealth

Based on the literature, we hypothesize a positive correlation between patience and income (and wealth), and we expect a negative relationship between present bias and income (and wealth).

In Fig 4 we do not observe strong relationships between time preferences and income. There is a marginal positive association (as expected) when omitting other variables (individuals with a $\delta = 1$ earning about 25, 000 Forints (76 EUR or 84.5 USD) more than their counterparts with a $\delta = 0.5$), but the correlation disappears as control variables are included in the regressions. Similarly, the coefficient of present bias is not significantly different from zero in any specification. For details, see Table D in S3 Appendix.

The correlation between patience and wealth is in line with our expectations. In Fig 5 we see a significant positive relationship when considered alone or even when we control for risk attitude. The association remains strong after controlling for age, gender and ethnicity and becomes marginally significant if we add regional and settlement type dummies. If we add

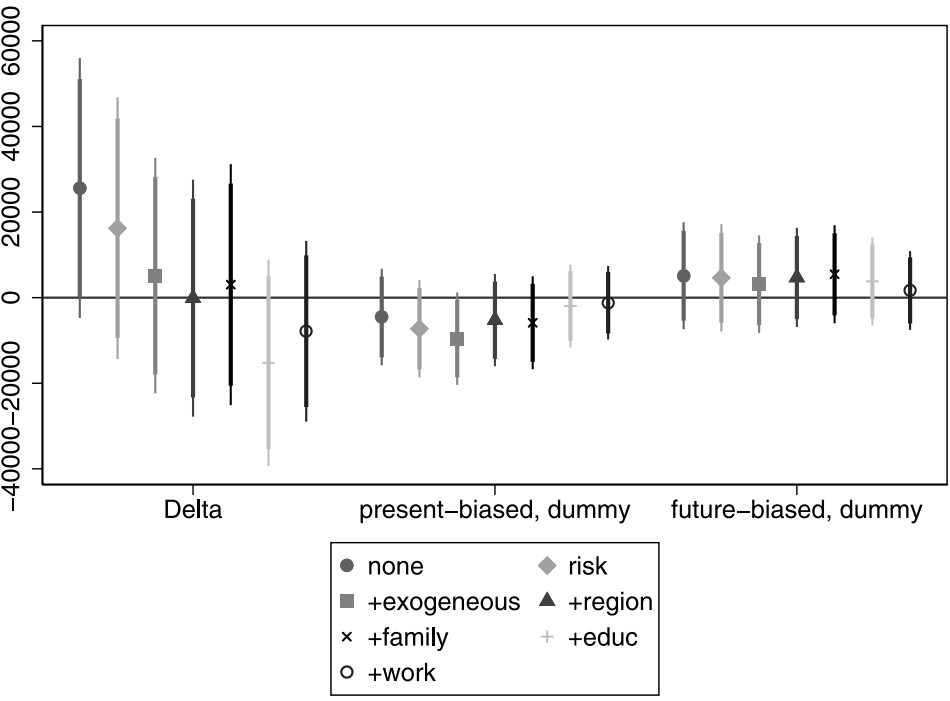

**Fig 4. The association of time preference with income.**

more control variables on *education* and *work*, the relationship ceases to be significant. However, these latter two, again, are probably bad controls as they directly impact wealth. Having this in mind, we would argue that patience could be a strong predictor of wealth. Present bias always has a negative sign as expected, and is also marginally significant, even if we take into

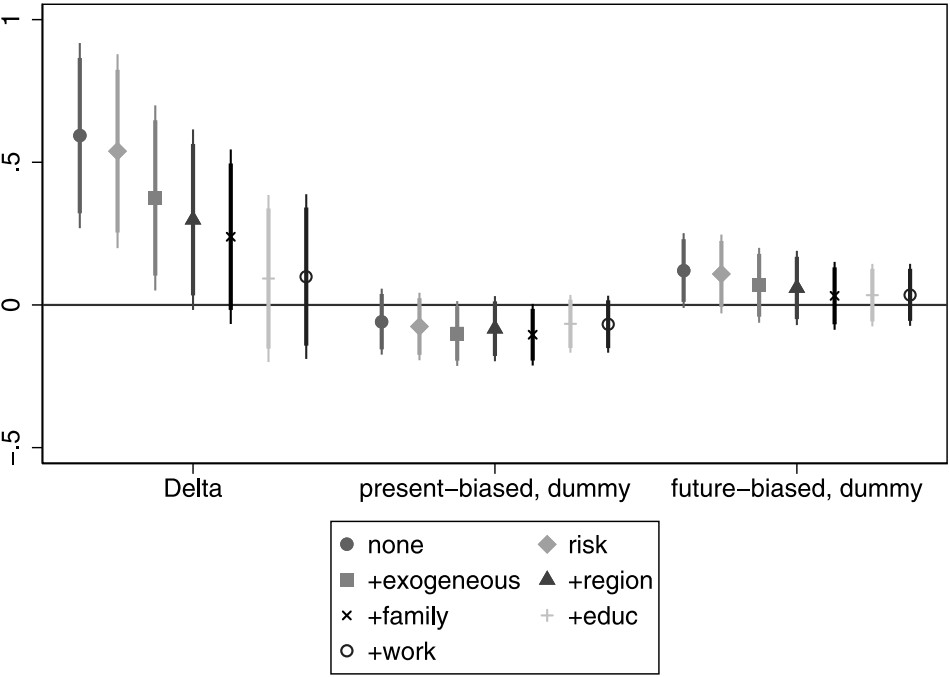

**Fig 5. The association of time preference with wealth.**

account variables related to *risk*, *exogenous*, *region* and *family*. This indicates that present-biased individuals tend to accumulate less wealth over their life. For details, see Table E in S3 Appendix.

**Finding 3 (income and wealth)**: Patience has mostly a positive, but generally insignificant association with income. For wealth the positive association is much more pronounced and significant in many specifications. The association between present bias and income and wealth is almost always negative (as expected), but this relationship is significant only marginally in some specifications, and mainly for wealth.

## Financial decisions and financial difficulties

**Banking and saving decisions.** Based on the literature, we expect that more patient individuals tend to i) use more the basic banking services, ii) save more. Regarding present bias, we expect to see the opposite associations.

In line with our hypothesis, we observe a steady positive association of patience with banking decisions in Fig 6. The significance subsides somewhat as we add more and more controls, but the association remains at least marginally significant even after including all controls. The present bias dummy is significantly and positively correlated with banking decisions in the first regressions. This positive association is not in line with our expectations. A potential explanation is that if an individual is aware of suffering from present bias, then she may use commitment devices to deal with this problem, see for instance [45]. Our data does not allow to check if this is the case here. For details, see Table F in S3 Appendix.

When looking at the individual components of the index, in the case of having a bank account, patience does not play a significant role, present bias shows a significant and positive relationship (in contrast to our conjecture) in the first regressions (present-biased individuals being 5.5 − 8 pp more likely to have a bank account), but not after controlling for *region*.

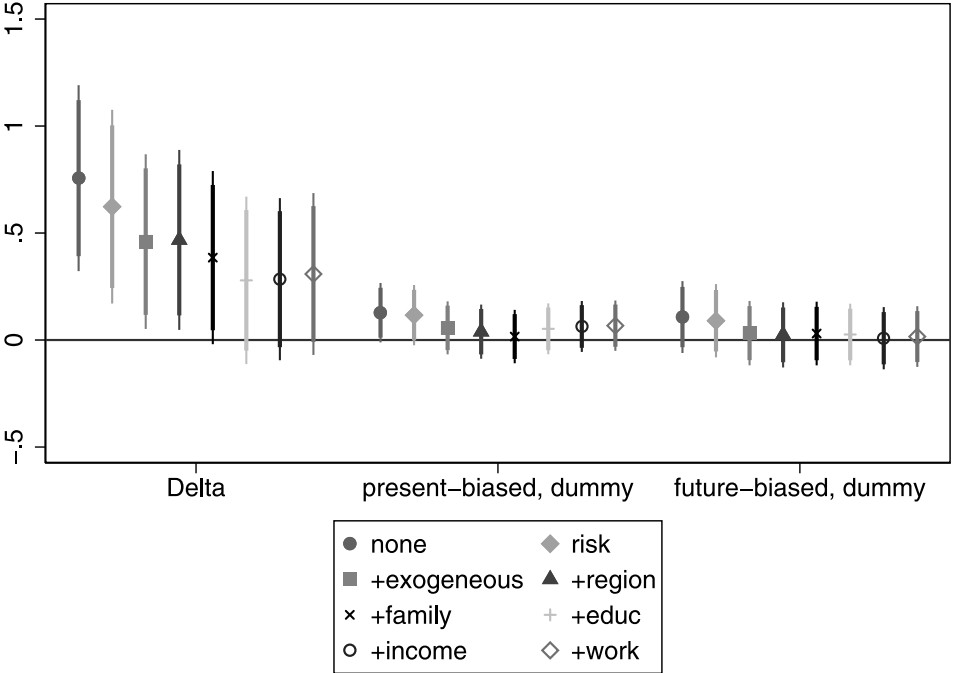

**Fig 6. The association of time preference with banking decisions.**

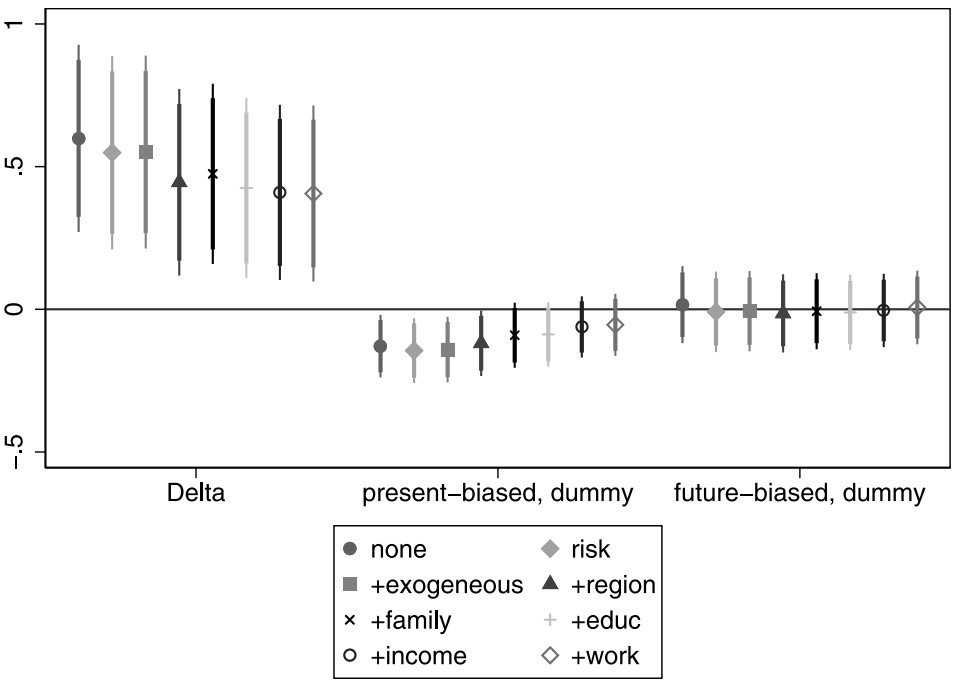

**Fig 7. The association of time preference with saving decisions.**

Concerning the ownership of debit / credit card, patience exhibits a consistent positive correlation (those with a $\delta = 1$ being about 13 pp more likely to have a debit / credit card) that is significant without controls, but the significance disappears after controlling for *exogenous* factors. Present bias also has a consistent positive association that is marginally significant without controls and when only risk tolerance is included as control. In these cases, present bias associates with a 5.5 pp higher probability of having a debit / credit card. For details, see Tables A and B in S4 Appendix.

When investigating saving decisions, in line with our hypotheses patience exhibits a significant positive association (see Fig 7), even if we include all controls. That is, more patient individuals are clearly more likely to have savings. Present bias also has the expected negative relationship and it is significant even if we control for risk attitude, *exogenous* and *region*. For details, see Table G in S3 Appendix.

Since we have an index, the interpretation of the coefficients is not straightforward, so we look at the components of the index. When considering separately the components of the saving index, we find that patience has a consistent and positive association with the ownership of stocks that is significant at least at 10% throughout the regressions. An individual with a $\delta = 1$ is about $3.5 - 5$ pp more likely to own a stock than an individual with a $\delta = 0.5$. The coefficient of present bias is also in line with the expectations, that is, present-biased individuals are 2 $- 3$ pp less likely to own stocks. This correlation is significant in all, but the last specifications. When turning to retirement savings, a similar but stronger pattern emerges. Patience shows a very strong positive association throughout the regressions. Even if we control for all the variables, the relationship is still significant at the 1% level. More concretely, individuals with a $\delta = 1$ are about $9 - 17$ pp more likely to have retirement savings than their peers with a $\delta = 0.5$. Present bias shows the expected negative association (present-biased individuals being 4 $- 7$ pp less likely to have retirement savings) and it is at least marginally significant in almost all specifications. Concerning holding a life insurance, patience has a consistent positive

association (as expected) that proves to be significant at least at 5% in all regressions. Individuals with a $\delta = 1$ are $12.5 - 22$ pp more likely to hold life insurance than individuals with a $\delta = 0.5$. However, present bias does not seem to play a role as it is not significant in any specification. For details, see Tables C, D and E in S4 Appendix.

**Finding 4 (financial decisions)**: Patience has a consistent positive and significant association with banking decisions, while present bias does not seem to play a role. When considering separately the components of banking decisions, having a bank account is not affected by patience and is positively and weakly influenced by present bias. Concerning having a debit / credit card, patience has a consistent positive association that is significant when some control variables are added. Present bias is negatively associated with having a debit / credit card, and the relationship is significant only when some control variables are added. Patience has a very strong positive and significant association with having savings. The association is strongest for retirement savings, followed by life insurance and owning stocks. We also observe that present-biased individuals are less likely to have savings, in general. This pattern is observed for stock ownership and retirement savings, but is absent in the case of life insurance.

**Financial difficulties.** According to Fig 8 patience is not associated with financial difficulties in our sample. The sign of the coefficient is consistently negative, suggesting that more patient individuals are less likely to suffer financial distress, but it fails to be significant in any of the specifications. However, present-biased individuals tend to have more financial difficulties. The coefficient of the present bias dummy is statistically significant in all specifications. This result is very similar to that found by [7] who study credit card borrowing and find that individual discount factors do not explain credit card debt, while present bias does. For details, see Table H in S3 Appendix.

**Finding 5 (financial difficulties)**: Patience associates in a consistently negative manner with the occurrence of financial difficulties (as expected), however the association fails to be

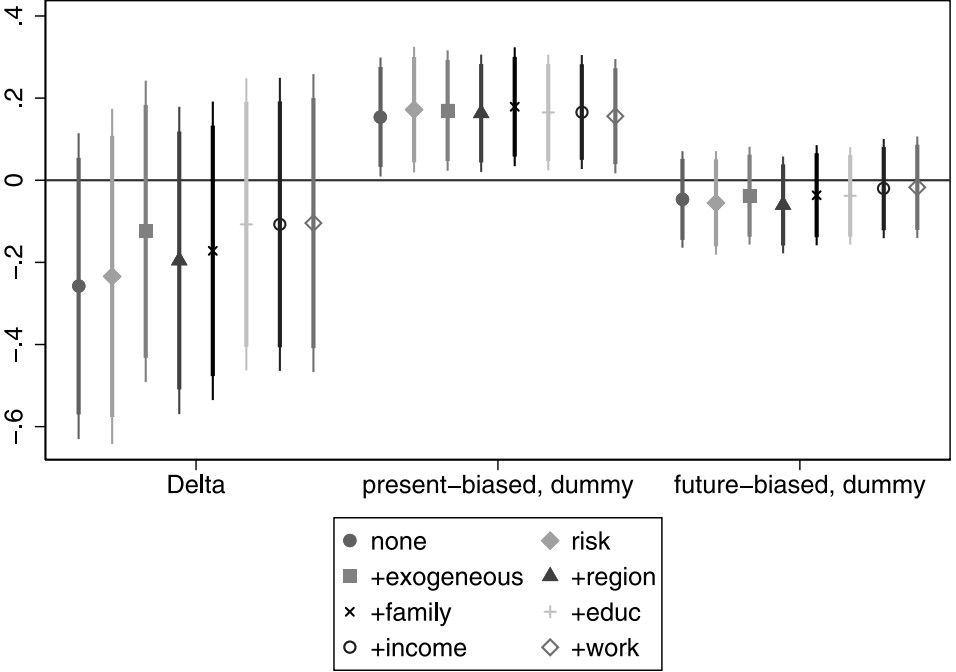

**Fig 8. The association of time preferences with financial difficulties.**

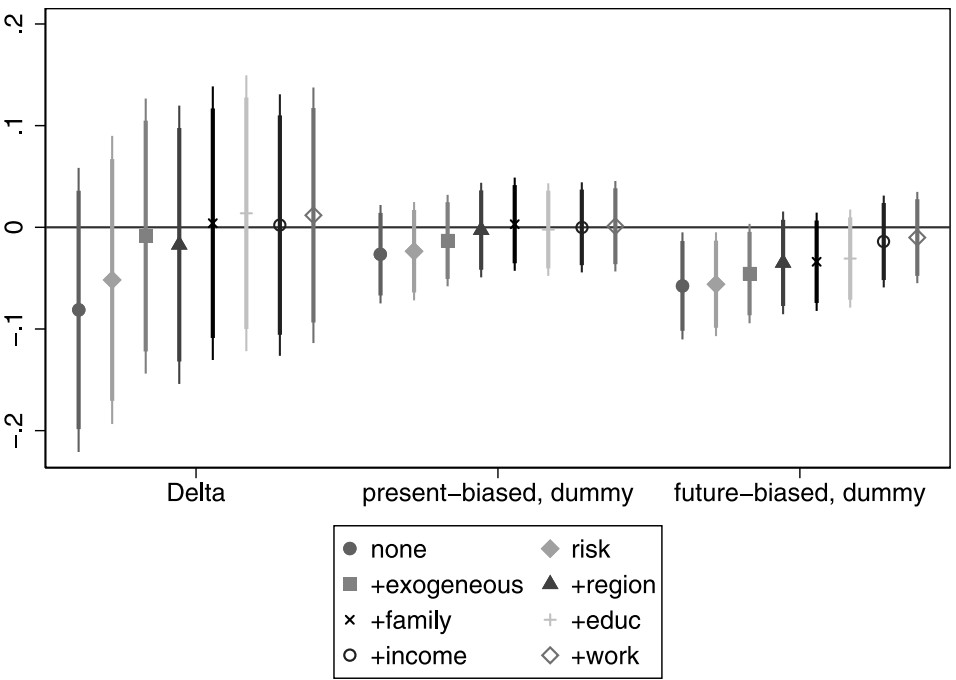

**Fig 9. The association of time preference with having bad health.**

significant. Present bias has the expected positive association with financial difficulties, moreover this relationship is significant even if we add all control variables.

## Health

Following [5], we form two dummy variables based on the answers given to the self-reported health question in the survey. If the respondent chooses one of the two categories that report an inferior health, then we say that the individual has bad health. Similarly, we construct a dummy variable standing for good health. Based on the literature, we expect a positive (negative) relationship between patience (present bias) and self-reported good health.

In Fig 9 we do not see any association of patience or present bias with bad health. Patience seems to have a generally negative association (as expected), but fails to be significant. The association of present bias is mainly negative (contrary to our expectations) and not significant in any of the specifications. Interestingly, being future-biased is significantly correlated with bad health in some specifications especially without controls: future-biased individuals are less likely to suffer bad health. For details, see Table I in S3 Appendix.

Fig 10 indicates that without control variables, patience associates significantly and positively with good health, as expected. The most patient individuals in our sample (those with a $\delta = 1$) are about 14 pp more likely to enjoy good health than those with a patience level of $\delta = 0.5$. The significance goes away as we add control variables, but reassuringly we see that the coefficient has a consistent positive sign. Present bias does not have a significant association, in line with [109]. For details, see Table J in S3 Appendix.

**Finding 6 (health)**: We find weak evidence that more patient individuals are more likely to enjoy good health, but we fail to document any significant association between patience and bad health or any correlation of present bias with self-reported health.

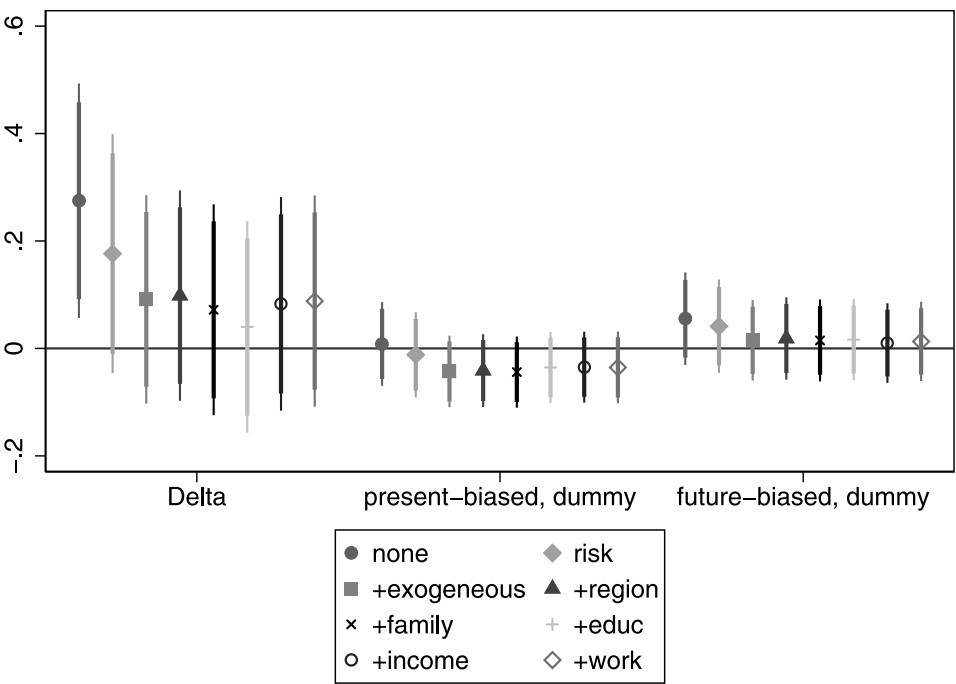

**Fig 10. The relationship between time preference and having good health.**

## Discussion

In this study we investigated the association between two aspects of time preference (patience and present bias) and life outcomes in five domains. Patience shows the expected association in most of the cases in terms of exhibiting consistently the expected sign and being significant in at least some specifications. The associations are the strongest with escaping low educational attainment, wealth, banking and saving decisions and financial difficulties. Only in the case of unemployment do we observe unexpected signs and associations.

Present bias seems to be less important, as it has low to no association with many outcomes. It has the expected sign and is significant in at least some specifications only when considering escaping the low educational outcome, banking and saving decisions, and financial difficulties. There are at least two potential explanations why the relationship between present bias and the domains under study is less pronounced than that of patience. First, if an individual is aware that she suffers from present bias, then she may use commitment devices [110] to overcome their problem as we already pointed out in the case of banking decisions. We cannot observe the use of commitment devices, so in principle we cannot discard the possibility that they have some effect. However, we do not see why commitment devices would help in the areas where we do not see an association between the variable of interest and present bias, but not in the other domains. Further studies should investigate, for instance, if commitment devices in financial services and issues could dampen the negative association of present bias. Second, [88]—among others—argue that present bias may be more marked in case of consumption goods than for monetary flows. We do not consider real consumption goods and actually we observe that present bias exhibits some significant associations in issues involving money (banking, saving and financial difficulties), so this explanation seems to have no importance in our case.

We consider a strength of our study that it uses a representative sample that allows us to draw firmer conclusions about the associations between time preferences and life outcomes. More such representative samples with more respondents under different elicitation methods and incentive schemes would be welcome so that we obtain a more robust knowledge on how patience and time inconsistency associate with relevant outcomes in life. Given a more conclusive knowledge we will able to come up with adequate policy interventions. Some steps have been made in that direction already. In line with the early childhood intervention literature (see for instance [111, 112]), [113] design an intervention to foster patience. The intervention indeed leads to more patient intertemporal choices in the case of the students treated and has a persistent positive effect on school performance. Our study (and others in this literature) suggest that such interventions may have an effect in other domains as well.

## Supporting information

**S1 Data.**
(DO)

**S2 Data.**
(DTA)

**S1 Appendix. Survey questions.**
(PDF)

**S2 Appendix. Staircase method to measure time preferences and individual discount rates.**
(PDF)

**S3 Appendix.**
(PDF)

**S4 Appendix. Financial decisions–separate regressions.**
(PDF)

## Acknowledgments

We are very grateful to Péter Biró, László Halpern, Marc Kaufmann, Balázs Kertész, László Lőrincz, Brigitta Németh and János Vince for enlightening comments, and thank participants of research seminar at the Institute of Economics of the Hungarian Academy of Sciences and the annual conference of the Hungarian Society of Economics for their helpful comments.

## Author Contributions

**Conceptualization:** Dániel Horn, Hubert János Kiss.

**Data curation:** Dániel Horn, Hubert János Kiss.

**Formal analysis:** Dániel Horn, Hubert János Kiss.

**Investigation:** Dániel Horn, Hubert János Kiss.

**Methodology:** Dániel Horn, Hubert János Kiss.

**Writing – original draft:** Dániel Horn, Hubert János Kiss.

**Writing – review & editing:** Dániel Horn, Hubert János Kiss.

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
