## [Decision Letter · Decision Letter 0]

29 Jan 2020

PONE-D-19-29612

Time preferences and their life outcome correlates: Evidence from a representative survey

PLOS ONE

Dear Dr. Kiss,

Thank you for submitting your manuscript to PLOS ONE. After careful consideration, we feel that it has merit but does not fully meet PLOS ONE’s publication criteria as it currently stands. Therefore, we invite you to submit a revised version of the manuscript that addresses the points raised during the review process.

I note here that novelty is not a publication criterion for PLOS ONE, which addresses some of the referees' comments. However, we evaluate research on the basis of scientific validity, rigorous methodology, and high ethical standards. In light of this, please make sure that your article does not use any language that inappropriately suggests causal relationships when your research design does not allow to identify causality. It would also be good if your discussion would address directly whether or not your results warrant a causal interpretation (e.g. in the context of possible policy implications you mentioned, which seem far less obvious without causal evidence). I would also suggest to avoid wording such as "we could not come up with a good story" and a careful editing of the article by a native speaker or a professional editing service. Please also address all other comments raised by the referees.

We would appreciate receiving your revised manuscript by Mar 14 2020 11:59PM. To enhance the reproducibility of your results, we recommend that if applicable you deposit your laboratory protocols in protocols.io, where a protocol can be assigned its own identifier (DOI) such that it can be cited independently in the future. For instructions see: http://journals.plos.org/plosone/s/submission-guidelines#loc-laboratory-protocols

We look forward to receiving your revised manuscript.

Kind regards,

Philipp D. Koellinger, Ph.D.

Academic Editor

PLOS ONE

Journal Requirements:

2. We note you collected primary data through inclusion in a third-party survey instrument with the TARKI Social Research Institute Omnibus Survey. Some issues regarding ethics and consent require your clarification:

a. Please clarify if the TARKI Omnibus survey has been evaluated for research ethics by an IRB or similar ethics board.

b. Please provide additional details regarding participant consent.

In the ethics statement in the Methods and online submission information, please ensure that you have specified (i) whether consent was informed and (ii) what type was obtained (for instance, written or verbal, and if verbal, how it was documented and witnessed).

If your study included minors, state whether you obtained consent from parents or guardians. If the need for consent was waived by the ethics committee, please include this information.”

3. We note you have included tables to which you do not refer in the text of your manuscript. Please ensure that you refer to Table 7, 8, 10 and 11 in your text; if accepted, production will need this reference to link the reader to each Table.

Reviewers' comments:

Reviewer's Responses to Questions

**Comments to the Author**

1. Is the manuscript technically sound, and do the data support the conclusions?

Reviewer #1: No

Reviewer #2: Yes

2. Has the statistical analysis been performed appropriately and rigorously? 

Reviewer #1: No

Reviewer #2: Yes

3. Have the authors made all data underlying the findings in their manuscript fully available?

Reviewer #1: Yes

Reviewer #2: No

4. Is the manuscript presented in an intelligible fashion and written in standard English?

Reviewer #1: Yes

Reviewer #2: Yes

5. Review Comments to the Author

Reviewer #1: This article studies the relationship between time preference and life outcomes in Hungary.

The article discusses an interesting theme and is well-written. I like that the paper uses such a rich set of controls (especially risk preferences) and that the authors separate the results for patience and time inconsistency. The literature review is impressive.

Main issue

The authors measure time preference in a cross-section and relate it to outcomes which are also recorded in this same cross-section. This implies that the results may be subject to reverse causality, i.e. educational attainment may for instance affect time preference instead of the other way around.

Other issues

The authors do not discuss which mechanisms may explain the correlations they find.

The contribution of the paper is not clear to me. There are several other papers which have studied the relationship between time preference and outcomes in representative data sets as highlighted by the authors. The results therefore do not provide new insights.

Many results were not expected by the authors. They hardly discuss reasons for these unexpected results.

A minor point of attention is that the authors sometimes write percent while they mean percentage points (see for instance P12).

Reviewer #2: Referee report on PONE-D-19-29612, entitled “Time preferences and their life outcome correlates: Evidence from a representative survey”

This paper studies the relation between time preferences and several different variables in a representative sample of the Hungarian adult population. The authors measure time preference assuming the quasi-hyperbolic discounting model, making a discounting between the amount of impatience (recoded to reflect the amount of patience) and the amount of present bias. The authors report the expected relation between time preference and most of their included outcomes, which is often significant for patience, but not so much for present bias. The study does not contain any methodological novelties and all these relations have been investigated before, but I understood that this is no main criterion for PLOS ONE, so I won’t get back to this issue here. In general, I think the study has been well-conducted, with an interesting dataset, and the manuscript has been well-written. My remaining comments are listed below.

1. The beta-delta discounting model is theoretically not valid for also capturing future bias. See, for instance, Bleichrodt et al. (2009).

2. Line 35: “contributions”.

3. Line 124: “also find”.

4. Line 125: “a relation”.

5. Lines 137-146: why do you use only two groups for education? In this way, you throw away information.

6. Line 163-164. This sentence is not very nice. Please rephrase.

7. Line 195: “a positive relationship”.

8. Line 227: “a significantly higher amount”.

9. Line 241: “trouble”.

10. Line 287: “a biased result” or “biased results”.

11. Line 289: Attema et al. (2018) also compared discounting for health and money and found different discount rates as well (more discounting for money than for health).

12. Lines 290-291. The word “larger” is redundant here.

13. Formulas also require punctuation.

14. Footnote 15: remove the comma after “the hypothesis that risk” in the third row.

15. Last sentence of footnote 15: this has also been shown by Abdellaoui et al. (2010).

16. Line 330: “what” should be “which”.

17. Line 335: “more” should be “less”.

18. Line 336: “then” should be “than”.

19. Second paragraph of page 10: I don’t get your point at the start of this paragraph. If patience and beta are negatively correlated, it appears to me that it would mean that those who are less patient, are LESS likely to suffer from present bias. This is because the higher beta, the lower will be the present bias.

20. Line 372: “present-biased”, not “present-biassed”.

21. On line 417 you write that not much is known about future bias. However, by now, there has been quite some studies either reporting a substantial amount of future bias or studying it from an analytical point of view (e.g. Attema et al. 2010, Takeuchi 2011, Bleichrodt et al. 2016, Attema and Lipman 2018, Rohde 2019, and reference therein).

22. Line 485: add a blank in front of “USD”.

23. Lines 496-497: “patience could be a strong predictor of wealth”. I think that reverse causality can also play a large role here, since those who are wealthy, probably don’t need the money now as much as poorer people.

24. Line 499: remove the blanks after “family”.

25. Lines 523-525: why do you consider the positive effect of present bias on debit/credit card ownership unexpected? One could also argue that people with a present bias are more likely to have credit card debts. By the way, it would be useful to make a distinction between credit and debit card ownership.

26. Lines 528-529: I think that for the relation between present bias and savings, the same caveat about reverse causality as in 23. applies.

References

1. Abdellaoui M., Attema A.E., Bleichrodt H. (2010). Intertemporal tradeoffs for gains and losses: An experimental measurement of discounted utility. The Economic Journal, 120 (545), 845-866.

2. Attema A.E., Bleichrodt H., L’Haridon O. (2018). Ambiguity preferences for health. Health Economics, 27 (11), 1699-1716.

3. Attema A.E., Bleichrodt H., Rohde K.I.M., Wakker P.P. (2010). Time-tradeoff sequences for analyzing discounting and time inconsistency. Management Science, 56 (11), 2015-2030.

4. Attema A.E., Lipman S.A. (2018). Decreasing impatience for health outcomes and its relation with healthy behavior. Frontiers in Applied Mathematics and Statistics, 4 (16).

5. Bleichrodt, H., Gao, Y., & Rohde, K. I. (2016). A measurement of decreasing impatience for health and money. Journal of Risk and Uncertainty, 52(3), 213-231.

6. Bleichrodt, H., Rohde, K. I., Wakker, P. P. (2009). Non-hyperbolic time inconsistency. Games and Economic Behavior, 66(1), 27-38.

7. Rohde, K. I. (2019). Measuring decreasing and increasing impatience. Management Science, 65(4), 1700-1716.

8. Takeuchi, K. (2011). Non-parametric test of time consistency: Present bias and future bias. Games and Economic Behavior, 71(2), 456-478.

6. PLOS authors have the option to publish the peer review history of their article (what does this mean?). If published, this will include your full peer review and any attached files.

Reviewer #1: No

Reviewer #2: No

---

## [Author Response · Author response to Decision Letter 0]

1 Jun 2020

We attached our responses to the reviewers and the editor.

---

## [Decision Letter · Decision Letter 1]

9 Jul 2020

Time preferences and their life outcome correlates: Evidence from a representative survey

PONE-D-19-29612R1

Dear Dr. Kiss,

We’re pleased to inform you that your manuscript has been judged scientifically suitable for publication and will be formally accepted for publication once it meets all outstanding technical requirements.

Kind regards,

Philipp D. Koellinger, Ph.D.

Academic Editor

PLOS ONE

Additional Editor Comments (optional):

Reviewers' comments:

Reviewer's Responses to Questions

**Comments to the Author**

1. If the authors have adequately addressed your comments raised in a previous round of review and you feel that this manuscript is now acceptable for publication, you may indicate that here to bypass the “Comments to the Author” section, enter your conflict of interest statement in the “Confidential to Editor” section, and submit your "Accept" recommendation.

Reviewer #1: All comments have been addressed

Reviewer #2: All comments have been addressed

2. Is the manuscript technically sound, and do the data support the conclusions?

Reviewer #1: Yes

Reviewer #2: Yes

3. Has the statistical analysis been performed appropriately and rigorously? 

Reviewer #1: Yes

Reviewer #2: Yes

4. Have the authors made all data underlying the findings in their manuscript fully available?

Reviewer #1: Yes

Reviewer #2: Yes

5. Is the manuscript presented in an intelligible fashion and written in standard English?

Reviewer #1: Yes

Reviewer #2: Yes

6. Review Comments to the Author

Reviewer #1: The authors have answered all my comments to my satisfaction. The paper has improved a lot. I have no further questions or comments.

Reviewer #2: (No Response)

7. PLOS authors have the option to publish the peer review history of their article (what does this mean?). If published, this will include your full peer review and any attached files.

Reviewer #1: No

Reviewer #2: No

---

## [Editor Report · Acceptance letter]

14 Jul 2020

PONE-D-19-29612R1 

Time preferences and their life outcome correlates: Evidence from a representative survey 

Dear Dr. Kiss:

I'm pleased to inform you that your manuscript has been deemed suitable for publication in PLOS ONE. Congratulations! Your manuscript is now with our production department. 

Kind regards, 

on behalf of

Dr. Philipp D. Koellinger 

Academic Editor

PLOS ONE